# Modified Albumin–Bilirubin Model for Stratifying Survival in Patients with Hepatocellular Carcinoma Receiving Anticancer Therapy

**DOI:** 10.3390/cancers14205083

**Published:** 2022-10-17

**Authors:** Wei-Fan Hsu, Shih-Chao Hsu, Te-Hong Chen, Chien-Hung Lin, Ying-Chun Lin, Yu-Wei Chang, Hung-Wei Wang, Yu-Min Liao, Hsueh-Chou Lai, Cheng-Yuan Peng

**Affiliations:** 1Center for Digestive Medicine, Department of Internal Medicine, China Medical University Hospital, Taichung 404327, Taiwan; 2Graduate Institute of Biomedical Sciences, China Medical University, Taichung 404327, Taiwan; 3School of Chinese Medicine, China Medical University, Taichung 404327, Taiwan; 4Department of Surgery, China Medical University Hospital, Taichung 404327, Taiwan; 5Department of Radiology, China Medical University Hospital, Taichung 404327, Taiwan; 6Department of Radiation Oncology, China Medical University Hospital, Taichung 404327, Taiwan; 7Department of Pathology, China Medical University Hospital, Taichung 404327, Taiwan; 8School of Medicine, China Medical University, Taichung 404327, Taiwan; 9Division of Hematology and Oncology, Department of Internal Medicine, China Medical University Hospital, Taichung 404327, Taiwan

**Keywords:** albumin–bilirubin (ALBI) grade, curative therapy, hepatocellular carcinoma, modified ALBI grade, survival

## Abstract

**Simple Summary:**

Albumin–bilirubin (ALBI) grade is an objective and reproducible model for evaluating overall survival (OS) in patients with hepatocellular carcinoma (HCC). The original ALBI grade was established for patients with Child–Pugh classes A–C. HCC patients with Child–Pugh class C or poor performance status usually receive hospice care. Thus, optimized cutoffs for the ALBI grade for stratifying OS in HCC patients receiving anticancer therapy are critical. The modified ALBI (mALBI) grade was as follows: an ALBI score ≤−3.02 for mALBI grade 1, an ALBI score >−3.02 to ≤−2.08 for mALBI grade 2, and an ALBI score > −2.08 for mALBI grade 3. The mALBI model can differentiate between patients with early, intermediate, or advanced HCC who received anticancer therapy.

**Abstract:**

Albumin–bilirubin (ALBI) grade is an objective and reproducible model for evaluating overall survival (OS) in patients with hepatocellular carcinoma (HCC). However, the original ALBI grade was established for patients with Child–Pugh classes A–C. HCC patients with Child–Pugh class C or poor performance status (Barcelona Clinic Liver Cancer (BCLC) stage D) usually receive hospice care. Thus, optimized cutoffs for the ALBI grade for stratifying OS in HCC patients receiving anticancer therapy are pertinent for accurate prognostication. This study retrospectively enrolled 2116 patients with BCLC stages A–C HCC after the exclusion of those ineligible for receiving anticancer therapy. The modified ALBI (mALBI) grades were: an ALBI score ≤−3.02 for mALBI grade 1, an ALBI score >−3.02 to ≤−2.08 for mALBI grade 2, and an ALBI score >−2.08 for mALBI grade 3. The original ALBI and mALBI grades were independent predictors of OS in all the enrolled patients and those receiving transarterial chemoembolization. In patients receiving curative therapy (radiofrequency ablation and surgical resection), the mALBI grade (grade 2 vs. 1 and grade 3 vs. 2) was an independent predictor of OS. Original ALBI grade 2 vs. 1 was an independent predictor of OS but not ALBI grade 3 vs. 2. The mALBI model can differentiate between patients with early, intermediate, or advanced HCC who received anticancer therapy into three prognostic groups. External validation of the proposed mALBI grade is warranted.

## 1. Introduction

Primary liver cancer was the third leading cause of cancer deaths and the sixth most common cancer in the world in 2020, and hepatocellular carcinoma (HCC) is the most common primary liver cancer [1]. East and Southeast Asia and sub-Saharan Africa are endemic areas of hepatitis B virus (HBV) or hepatitis C virus (HCV), and the incidence of HCC is higher in these areas [2]. Chronic liver disease or cirrhosis is a major risk factor of HCC. The Barcelona Clinic Liver Cancer (BCLC) staging system is currently the gold standard for guiding therapies and predicting outcomes [3]. The Child–Pugh classification, which describes the main clinical presentations of liver decompensation [4], is an essential component of the BCLC staging system. In addition to tumor burden, the prognosis of a patient with HCC depends on their liver functional reserve [5].

The albumin–bilirubin (ALBI) grade, a simple model developed in 2015, includes only the serum total bilirubin and albumin levels and has been used to evaluate liver function in patients with HCC [6]. Due to its objectiveness, reproducibility, and simplicity, the ALBI grade is widely used to evaluate the prognosis of patients with HCC [7]. The ALBI grade and ALBI-based models have been used to predict the overall survival (OS) in patients receiving various therapies, such as radiofrequency ablation (RFA) [8,9], surgical resection [5,8,10], transarterial chemoembolization (TACE) [5,11], and systemic therapy [12,13]. Several modifications have been proposed to improve the discriminative performance of the ALBI grade, including the ALBI-T score [14] and the categorization of ALBI grade 2 into two subgroups (2a and 2b) [15].

The original ALBI grade was validated in patients with Child–Pugh classes A–C [6]. In clinical practice, HCC patients with Child–Pugh class C or a poor performance status (BCLC stage D) usually receive hospice care [3]. Thus, optimized ALBI grade cutoffs for stratifying OS in patients with HCC receiving anticancer therapy are pertinent for accurate prognostication. This study identified modified ALBI grade cutoffs for patients with early, intermediate, or advanced HCC receiving anticancer therapy.

## 2. Materials and Methods

### 2.1. Patients

A total of 2819 consecutive patients with newly diagnosed HCC visited China Medical University Hospital (CMUH) from January 2012 to October 2020, and 2743 patients were eligible for investigation after the exclusion of patients without a documented ALBI score (*n* = 76). In addition, patients who underwent liver transplantation (*n* = 143), received palliative care (*n* = 350), were transferred to other hospitals (*n* = 89), or had BCLC stage D (*n* = 239) were excluded. Some patients with BCLC stage A–C HCC received palliative care because of old age, seeking alternative therapy, or refusing therapy. Patients might have met more than one exclusion criterion. Finally, 2116 patients were enrolled in the analysis (Figure 1).

Demographic parameters, complete blood count, biochemical data, HCC etiologies, diabetes mellitus (DM) and liver cirrhosis (LC) status, and performance status were recorded at baseline. Tumors were examined through contrast-enhanced dynamic computed tomography (CT) or magnetic resonance imaging (MRI). This study followed the 1975 Declaration of Helsinki, and the Research Ethics Committee of CMUH approved this study (CMUH108-REC3-140). The need for informed consent was waived, because each personal identification was encrypted for privacy protection.

### 2.2. Diagnosis and Laboratory Tests

HCC was diagnosed based on pathological findings or typical radiological presentations of at least two imaging investigations, including abdominal ultrasonography, hepatic arterial angiography, contrast-enhanced dynamic CT, and MRI [16,17]. The Eastern Cooperative Oncology Group performance scale was used to evaluate the performance status [18]. Complete blood counts (Sysmex, Kanagawa, Japan) and blood biochemistry data (Beckman Coulter, Brea, CA, USA) were analyzed at the hospital’s central laboratory. HBV infection and HCV infection were defined as the presence of the serum hepatitis B surface antigen and the serum anti-HCV antibody plus detectable HCV RNA, respectively (Roche Diagnostics, Branchburg, NJ, USA). LC was determined based on the presence of an unequivocal clinical, ultrasonographic, or histological analysis.

### 2.3. Formula for ALBI Score and Modified ALBI Grade

Patients’ ALBI scores [6] were calculated using the following formula:ALBI score = log10 bilirubin (μmol/L) × 0.66 − albumin (g/L) × 0.085

In the original investigation, patients with HCC were classified into low-, medium-, and high-risk groups according to OS, which was stratified by splitting the linear predictors of the ALBI scores at the 25th and 90th percentiles (an ALBI score ≤−2.60 for grade 1, an ALBI score >−2.60 to ≤−1.39 for grade 2, and an ALBI score >−1.39 for grade 3) [6].

This study classified patients with HCC who received therapeutic interventions into three prognostic groups. Since only 3.1% (66/2116) of the enrolled patients belonged to the original ALBI grade 3, we modified the ALBI grade based on OS stratified by splitting the linear predictors of ALBI scores at different percentiles; optimization was performed using a multivariable Cox regression analysis.

### 2.4. Treatment

The therapeutic decisions of patients with HCC were discussed, and treatment was recommended by the multidisciplinary liver cancer team at CMUH. On the basis of the guidelines of the American Association for the Study of Liver Diseases and the European Association for the Study of the Liver [16,17], the Liver Cancer Clinical Practice Guidelines of CMUH were established to direct clinical practice. The indications for and extent of RFA and surgical resection were, respectively, determined by experienced gastroenterologists and surgeons at our hospital, as described previously [19,20]. RFA and surgical resection for HCC were considered curative therapies. The indications of TACE were patients with a fair liver reserve (a Child–Pugh score ≤8) and patent main portal vein or main portal vein thrombosis with collateral circulation. Radiotherapy is an alternative noninvasive ablative treatment for unresectable HCC with a high local control rate [21]. However, because only 119 patients received first-line radiotherapy, we did not perform a subgroup analysis of such patients. The procedures of TACE and radiotherapy have been described in detail elsewhere [22,23]. Systemic therapy included tyrosine kinase inhibitors (TKIs, sorafenib and lenvatinib, *n* = 139); immune checkpoint inhibitors (ICIs, including nivolumab, pembrolizumab, and atezolizumab plus bevacizumab, *n* = 15); and systemic chemotherapy (carboplatin- and gemcitabine-based regimens, *n* = 5). The doses of the systemic therapies were prescribed in accordance with the recommended protocols [24].

### 2.5. Statistical Analysis

Continuous variables were presented as medians (1st quartile to 3rd quartile), and categorical variables were presented as frequencies (percentages). Data were censored until death, loss, or the end of follow-up (until 28 February 2022), whichever occurred first. Mann–Whitney *U* test was used to compare continuous variables between groups. Variables with a *p*-value of <0.20 in the univariate analysis were subjected to a multivariable Cox regression analysis to examine their associations with the OS. According to the OS, the enrolled patients (*n* = 2116) were classified into low-, medium-, and high-risk groups; stratification was performed by splitting the linear cutoffs of the ALBI scores, which were optimized by performing a multivariable Cox regression analysis using the enter method.

The discriminatory performance of the original ALBI grade and modified ALBI (mALBI) grade were calculated using Harrell’s C statistic and determined visually by performing a Kaplan–Meier analysis. The correlation between the Child–Pugh score or class and the original ALBI grade or the mALBI grade was examined using Somers’ delta [6]. The OS among the patient groups were compared by the Kaplan–Meier analysis with the log–rank test.

The bootstrap method with 2000 replications was used for the internal validation of the mALBI grade to estimate the 95% confidence interval (CI) for the coefficients of ALBI and mALBI grades determined through the Cox regression analyses. A 95% CI that included zero for the target variable indicated statistical nonsignificance. Details on the procedure were described previously [25]. In brief, the bootstrap method is a random resampling method with replacement and is used to estimate the distribution of target variables; it has advantages of convenience and efficiency for model development and internal validation [25,26].

Statistical analyses were performed using SPSS (version 25.0, IBM, New York, NY, USA), and Harrell’s C was calculated using MedCalc (Version 20.112, MedCalc Software Ltd., Ostend, Belgium). A two-sided *p*-value of <0.05 was considered statistically significant.

## 3. Results

### 3.1. Baseline Characteristics

Of the 2743 patients eligible for investigation, 627 were excluded. The excluded patients had higher aspartate aminotransferase (AST), alanine aminotransferase (ALT), total bilirubin, and α-fetoprotein (AFP) levels; a higher international normalized ratio (INR); a lower platelet count; a lower albumin level; a lower proportion of HCV infection; a higher proportion of Child–Pugh class B or C; higher fibrosis-4 (FIB-4), ALBI, Model for End-Stage Liver Disease (MELD), and Cancer of the Liver Italian Program (CLIP) scores; a higher BCLC stage and larger tumor size; and a shorter follow-up duration and median OS than the enrolled patients. Furthermore, the excluded patients were more likely to have LC. Most of the excluded patients received palliative care or were transferred to other hospitals (Appendix A).

Of the 2116 enrolled patients, 1513 (71.5%) were men, and 1227 (58.0%) had LC; the median age was 64 (56–72) years, and the median follow-up duration was 23.23 (8.22–51.08) months. Furthermore, 968 (45.7%), 590 (27.9%), and 558 (26.4%) had BCLC stages A, B, and C, respectively. Among the enrolled patients, 428 (20.2%), 781 (36.9%), 1073 (50.7%), and 847 (40.0%) reported alcohol consumption, having DM, having HBV infection, and having HCV infection, respectively. Most patients were categorized into Child–Pugh class A (*n* = 1808, 85.4%). The median FIB-4 and ALBI scores were 3.22 (1.97–5.72) and −2.66 (−3.02 to −2.20), respectively. The median maximum tumor size was 3.8 (2.2–7.5) cm, and fewer patients had a tumor volume that was >50% of the liver volume (*n* = 302, 14.3%). The median OS was 44.94 (95% CI: 39.54–50.34) months. The median AFP level was 30.24 (5.70–491.17) ng/mL. Patients with LC had higher AST, ALT, and total bilirubin levels; a higher INR; a higher proportion of HCV infection, DM, Child–Pugh class B, and HCC with macrovascular invasion (MVI); a lower platelet count; lower albumin and creatinine levels; a lower proportion of men and HBV infection; and a shorter follow-up duration and median OS (35.48 months vs. not reached, *p* < 0.001) than those without LC. Moreover, the patients with LC had higher FIB-4, ALBI, MELD, and CLIP scores. A higher percentage of patients without LC (440/889, 49.5%) received surgical intervention, and a higher percentage of patients with LC received RFA (307/1227, 25.0%) and TACE (549/1227, 44.7%; Table 1).

### 3.2. Comparison with the Original Study and Derivation of the mALBI Grade

In the patients eligible for investigation (*n* = 2743), the median age, total bilirubin level, and sex distribution were similar to those in the Japanese training set of the original ALBI study [6]. Our patients had higher albumin levels, and a higher proportion of them had Child–Pugh class A or MVI. The tumor characteristics in the original study were unknown [6]. The patients in the Japanese training set had the longest median OS, and the median OS were 26.0 and 7.2 months in the Spanish and Chinese cohorts, respectively, in the original study [6]. The cutoffs were −2.94 and −1.38 at the 25th and 90th percentiles, respectively, in the patients eligible for investigation in this study (Table 2).

We stratified the enrolled patients (*n* = 2116) into three groups by splitting the linear cutoffs of the ALBI scores at the 25th and 80th percentiles; these data were optimized by performing a multivariable Cox regression analysis with the enter method. The mALBI grade was defined as follows: the ALBI score was ≤−3.02 for mALBI grade 1, >−3.02 to ≤−2.08 for mALBI grade 2, and >−2.08 for mALBI grade 3 (Table 2).

### 3.3. mALBI Grade Instead of the Original ALBI Grade as a Predictor of OS in Patients Receiving Curative Therapy

We explored whether the mALBI grade could serve as a predictor of survival. Male sex, ALT (>40 U/L), AFP (≥400 ng/mL), tumor size (cm), MVI, ALBI grade, and mALBI grade were significantly associated factors of survival in the univariate Cox regression analysis. In the first multivariable Cox regression analysis model using the original ABLI grade (Model 1), AFP (≥400 ng/mL), tumor size, MVI, and ALBI grade (hazard ratio (HR): 2.370, 95% CI: 2.074–2.708, *p* < 0.001 for grade 2 vs. 1; HR 1.771, 95% CI: 1.323–2.370, *p* < 0.001 for grade 3 vs. 2) were independent predictors of OS in the enrolled patients. In the second multivariable Cox regression analysis model using the mALBI grade (Model 2), AFP (≥400 ng/mL), tumor size, MVI, and mALBI grade (HR: 1.881, 95% CI: 1.568–2.257, *p* < 0.001 for grade 2 vs. 1; HR 2.126, 95% CI: 1.844–2.452, *p* < 0.001 for grade 3 vs. 2) were independent predictors of OS in the enrolled patients (Table 3). In the patients with BCLC stage A (*n* = 968), mALBI grade (grade 2 vs. 1 and grade 3 vs. 2) was an independent predictor of OS in Model 2; however, the difference between ALBI grade 3 vs. 2 did not reach statistical significance (*p* = 0.127) in Model 1 (Appendix A). In patients with BCLC stage B (*n* = 590) and BCLC stage C (*n* = 558), ALBI grade (Model 1) and mALBI grade (Model 2) were the independent predictors of OS in the multivariable Cox regression analysis (Appendix A).

In the patients receiving curative therapy (RFA and surgical resection, *n* = 1042), the platelet count (≥100 × 10^9^/L), AFP (≥400 ng/mL), tumor size, MVI, and mALBI grade (grade 2 vs. 1 and grade 3 vs. 2) were independent predictors of OS in Model 2. Besides the platelet count (≥100 × 10^9^/L) and tumor factors (AFP, the tumor size, MVI), ALBI grade 2 vs. 1 were independent predictors of OS in Model 1. However, ALBI grade 3 vs. 2 did not reach statistical significance (*p* = 0.085; Table 4). Similarly, in the patients undergoing surgical resection (*n* = 629), the mALBI grade (grade 2 vs. 1 and grade 3 vs. 2) was an independent predictor of OS in Model 2 but ALBI grade 3 vs. 2 did not reach statistical significance (*p* = 0.086) in Model 1 (Appendix A). In the patients receiving RFA (*n* = 413), platelet count (≥100 × 10^9^/L), AFP (≥400 ng/mL), tumor size, ALBI grade 2 vs. 1 (Model 1), and mALBI grade 3 vs. 2 (Model 2) were independent predictors of OS in the multivariable Cox regression analysis; however, ALBI grade 3 vs. 2 (*p* = 0.187 in Model 1) and mALBI grade 2 vs. 1 (*p* = 0.117 in Model 2) did not reach statistical significance (Appendix A).

In the patients receiving TACE (*n* = 799), the platelet count (≥100 × 10^9^/L), AFP (≥400 ng/mL), tumor size, MVI, ALBI grade (Model 1), and mALBI grade (Model 2) were independent predictors of OS in the multivariable Cox regression analysis (Appendix A). In the patients receiving systemic therapy (*n* = 159), the tumor size, ALBI grade 3 vs. 2 (Model 1), and mALBI grade 3 vs. 2 (Model 2) were independent predictors of the OS; however, ALBI grade 2 vs. 1 (*p* = 0.057 in Model 1) and mALBI grade 2 vs. 1 (*p* = 0.309 in Model 2) did not reach statistical significance (Appendix A).

In the patients categorized as Child–Pugh class A (*n* = 1808), male sex, ALT (>40 U/L), tumor factors, ALBI grade (*p* < 0.001 for grade 2 vs. 1; *p* = 0.051 for grade 3 vs. 2), and mALBI grade were significantly associated with the OS. In the Cox regression analysis, the tumor factors, ALBI grade (Model 1), and mALBI grade (Model 2) were independent predictors of OS in patients categorized as Child–Pugh class A (Appendix A).

### 3.4. Discriminatory Performance of mALBI Grade and Original ALBI Grade

The Kaplan–Meier analysis results revealed that the probability of survival significantly differed among the patients with different mALBI grades (Figure 2A) and ALBI grades (Figure 2B), among those receiving curative therapies with different mALBI grades (Figure 2C) and ALBI grade 2 vs. 1 (Figure 2D), and among those receiving TACE with different mALBI grades (Figure 2E) and ALBI grades (Figure 2F). The probability of survival in the patients receiving curative therapy between the original ALBI grade 3 and 2 did not differ significantly (*p* = 0.064, Figure 2D). The values of Harrell’s C statistic were similar for the mALBI grade (0.640, 95% CI: 0.625–0.656, Figure 2A) and the original ALBI grade (0.643, 95% CI: 0.628–0.658, Figure 2B) among all enrolled patients. The values of Harrell’s C statistic for the mALBI and ALBI grades were similar in patients receiving curative therapy (Figure 2C,D) and TACE (Figure 2E,F).

The findings of the Kaplan–Meier analysis for the probability of survival in the patients with BCLC stage A (Appendix A), B (Appendix A), or C (Appendix A); those receiving surgical resection (Appendix A), RFA (Appendix A), or systemic therapy (Appendix A); and those with Child–Pugh class A (Appendix A) are presented in the Appendix A. The values of Harrell’s C statistic for mALBI and ALBI grades were similar in the subgroup analyses (Appendix A). Notably, the probability of survival between patients with ALBI grade 3 vs. 2 and BCLC stage A (*p* = 0.125, Appendix A), between patients with ALBI grade 3 vs. 2 and BCLC stage C (*p* = 0.076, Appendix A), and between patients with ALBI grade 3 vs. 2 who were receiving surgical resection (*p* = 0.275, Appendix A) and RFA (*p* = 0.111, Appendix A) did not differ significantly.

Visual inspection of the Kaplan–Meier plots revealed that the mALBI grade had better ability to discriminate the three prognostic groups in the entire group of enrolled patients or those receiving curative therapy, surgical resection, RFA, or TACE compared with the original ALBI grade (Figure 2A–F and Appendix A).

The correlations between the Child–Pugh score or class and mALBI grade or original ALBI grade were examined using Somers’ delta. The values of Somers’ delta were higher between Child–Pugh score or class and mALBI grade than in those between Child–Pugh score or class and the original ALBI grade (Appendix A).

### 3.5. Internal Validation Using the Bootstrap Method

The bootstrap method for internal validation was used to estimate the 95% CIs for the coefficients of the ALBI and mALBI grades. The 95% CI for the coefficients of the ALBI (Model 1) and mALBI (Model 2) grades did not include zero in all the enrolled patients. The 95% CI for the coefficients of the mALBI grade (Model 2) did not cross zero, but the 95% CI for coefficient of ALBI grade 3 vs. 2 (Model 1) crossed zero without statistical significance (B, 0.573, 95% CI: −0.206 to −1.329, *p* = 0.120) for the patients receiving curative therapy. The 95% CI for the coefficients of ALBI grade 3 vs. 2 (Model 1, B 1.324, 95% CI: −0.056 to −2.470, *p* = 0.009) and mALBI grade 2 vs. 1 (Model 2, B 0.399, 95% CI: −0.260 to −1.165, *p* = 0.194) crossed zero for patients receiving systemic therapy (Appendix A).

## 4. Discussion

According to the OS, the mALBI model could stratify patients with HCC receiving anticancer therapy into three groups. The mALBI model could be applied to all the enrolled patients; those with BCLC stage A (*n* = 968), *B* (*n* = 590), and C (*n* = 558); those receiving curative therapies (RFA and surgical resection, *n* = 1042) or TACE (*n* = 799); and those with Child–Pugh class A (*n* = 1808). Instead of the original ALBI grade, the mALBI grade was a predictor of OS in patients with BCLC stage A and those receiving curative therapy (Figure 2C,D, and Table 4) or surgical resection (Appendix A). Therefore, the modified cutoffs for mALBI grade—ALBI score ≤−3.02 (grade 1), an ALBI score >−3.02 to ≤−2.08 (grade 2), and an ALBI score >−2.08 (grade 3)—are more suited for stratifying patients with HCC receiving anticancer therapy compared with the original cutoffs for the ALBI grade.

HCC is a critical health problem globally, and current therapies have limitations. Surgical resection is the treatment of choice in selected patients; however, a significant proportion (as high as 70% at 5 years) of patients experience recurrence [27]. In patients with BCLC stage 0 or A who are not eligible for liver transplantation or surgical resection, RFA is an alternative curative approach. The median time to recurrence after RFA is 20–30 months, and the 5-year recurrence rate is approximately 50–70% [28,29]. TACE significantly prolonged the OS in patients with intermediate HCC [30], but the median OS ranged from 20 to 37 months in randomized controlled trials [31]. Therefore, the development of a simple and reproducible model to predict survival in patients with HCC is crucial.

In a cohort of 2559 patients with HCC receiving curative therapy (more than 90% of patients belonged to Child–Pugh class A), patients with ALBI grade 1 had a longer OS than those with ALBI grade 2 [8]. Ho et al., demonstrated that the ALBI grade was a better predictor of OS than other liver function models, including Child–Pugh class and FIB-4, in patients with HCC undergoing RFA [9]. Lee et al., developed the ALBI–TAE model, which included the ALBI grade, AFP level, and up to 11 criteria, to predict the OS of patients with intermediate HCC undergoing TACE [11]. Our previous study revealed that the ALBI grade exhibited a high predictive value for OS in patients with intermediate HCC [5].

The current ALBI grade has some limitations. Patients were usually categorized into two groups instead of three groups as originally proposed for the survival analysis, such as patients with ALBI grade 2 or 3 vs. those with ALBI grade 1 [9,11], because only a limited number of patients with HCC with ALBI grade 3 received therapeutic interventions. Consistent with this clinical practice, only 3.1% of the enrolled patients in this study had an original ALBI grade 3. To overcome this limitation, Hiraoka et al., used the indocyanine green retention test (<30%) as a reference to propose a model to divide ALBI grade 2 into two subgrades (2a and 2b) [15]. In the present study, we determined the optimal values for splitting the linear cutoffs of the ALBI scores by performing a multivariable Cox regression analysis and proposed the use of the mALBI grade to stratify patients with early, intermediate, or advanced HCC based on the OS after excluding those ineligible for receiving anticancer therapy. We demonstrated that, compared with the ALBI model, the mALBI model better differentiated between patients with HCC receiving anticancer therapy into three groups.

TKIs and ICIs are the current frontline therapy for advanced HCC [3]. We previously demonstrated that the combination of the ALBI grade (2 vs. 1) and AFP level (≥20 ng/mL) could stratify patients with unresectable HCC (uHCC) for sorafenib–regorafenib sequential therapy [32]. Ueshima et al., demonstrated that patients with ALBI grade 1 and a Child–Pugh score of 5 had the highest objective response rate (57.1%) among patients with uHCC treated with lenvatinib [33]. Lee et al., established an ALBI-based integrated model, termed ALBI-progressive disease (PD), to select patients with better OS or PD after sorafenib therapy [12]. Lee et al., revealed the ALBI grade (2 or 3 vs. 1) to be an independent factor associated with OS in patients with ICI-treated uHCC (96.8% and 3.2% patients received nivolumab and pembrolizumab, respectively) [34]. Similarly, these studies have categorized patients into two groups instead of three groups (grade 3 vs. 1 or 2 [12] and grade 2 or 3 vs. 1 [34]). In this study, we enrolled patients until October 2020, and only 139 and 15 patients received first-line TKI and ICI therapy, respectively. The OS did not significantly differ between patients with ALBI grades 1 and 2 (*p* = 0.057) and those with mALBI grades 1 and 2 (*p* = 0.309, Appendix A), which could be attributed to the small number of enrolled patients receiving systemic therapy.

This study has several limitations. First, this was a single-center retrospective study, and HBV (50.7%) is the most common etiology of chronic liver disease. Although we used the bootstrap method for internal validation, the meticulous external validation of the mALBI grade in patients with HCC of other etiologies from other geographical regions is warranted. Second, referral bias cannot be completely excluded, because our center is a referral medical center. Whether the severity of liver disease in the present cohort is representative of the full spectrum of the hepatic reserve in patients with HCC receiving anticancer therapy requires further validation. Third, TKIs and ICIs are currently the frontline therapy for advanced HCC. Only 139 and 15 patients received first-line systemic therapy of TKIs and ICIs, respectively, in this study. The application of the mALBI grade for stratifying survival in patients treated with systemic therapy should be validated in future studies. Fourth, because the indocyanine green retention test was not routinely performed in patients with HCC in this study, it remains to be studied whether categorization of the mALBI grade 2 into 2a vs. 2b using the indocyanine green retention test as a reference can further improve the prognostic performance of the mALBI model. Last, the effect of temporal changes in the mALBI grade during HCC progression on treatment options and OS survival warrants further investigation.

## 5. Conclusions

The mALBI model can differentiate between patients with early, intermediate, or advanced HCC who received anticancer therapy into three prognostic groups. The mALBI model could be applied in different clinical settings, including for patients receiving curative therapy or TACE. However, the external validation of the mALBI model is warranted.

## Figures and Tables

**Figure 1 cancers-14-05083-f001:**
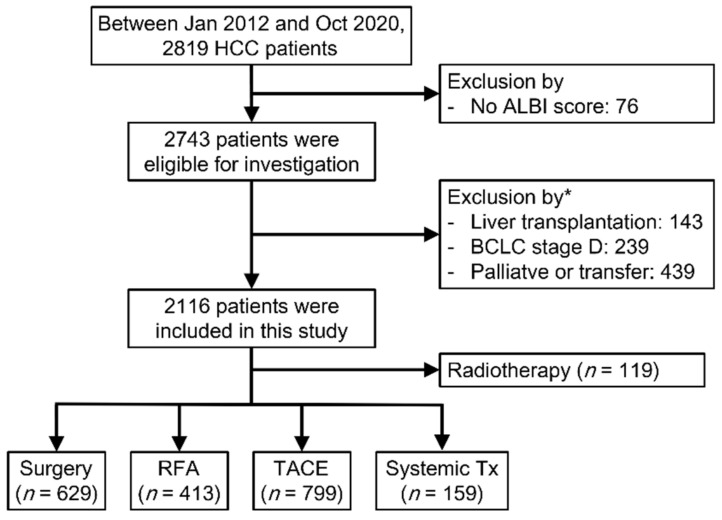
Flowchart of patient enrollment and stratification. ALBI, albumin–bilirubin; BCLC, Barcelona Clinic Liver Cancer; HCC, hepatocellular carcinoma; RFA, radiofrequency ablation; TACE, transarterial chemoembolization. * Patients might have met more than one exclusion criterion.

**Figure 2 cancers-14-05083-f002:**
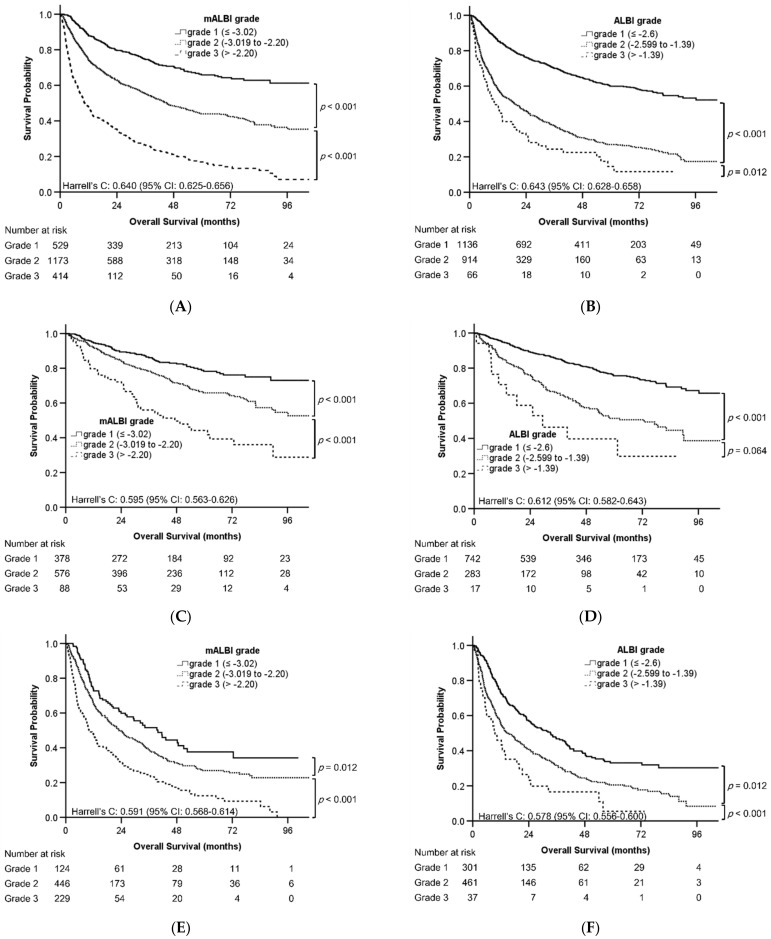
Kaplan–Meier analyses of the overall survival. (**A**) mALBI grade of all patients. (**B**) Original ALBI grade of all patients. (**C**) mALBI grade of patients receiving curative therapy. (**D**) Original ALBI grade of patients receiving curative therapy. (**E**) mALBI grade of patients receiving TACE. (**F**) Original ALBI grade of patients receiving TACE. ALBI, albumin–bilirubin; CI, confidence interval; mALBI, modified ALBI; TACE, transarterial chemoembolization.

**Table 1 cancers-14-05083-t001:** Patient demographics, baseline characteristics, and outcomes.

*n* (%) or Median (IQR)	Total (*n* = 2116)	Non-Cirrhosis (*n* = 889)	Cirrhosis (*n* = 1227)	*p*-Value
Age (years)	64 (56–72)	64 (54–72)	65 (57–72)	0.088
Sex (male), *n* (%)	1513 (71.5)	667 (75.0)	846 (68.9)	0.002
Follow-up months	23.23 (8.22–51.08)	25.92 (10.20–55.00)	21.49 (7.00–48.59)	<0.001
Platelet count (×10^9^/L)	157 (110–214)	184 (143–244)	137 (90–186)	<0.001
AST (U/L)	49 (32–84)	42 (29–71)	54 (35–91)	<0.001
ALT (U/L)	41 (26–64)	38 (25–58)	43 (27–68)	<0.001
Total bilirubin (mg/dL)	0.9 (0.6–1.2)	0.8 (0.6–1.1)	1.0 (0.7–1.4)	<0.001
Albumin (g/dL)	4.0 (3.6–4.4)	4.2 (3.8–4.5)	3.9 (3.4–4.3)	<0.001
INR	1.08 (1.02–1.14)	1.05 (1.00–1.11)	1.10 (1.04–1.17)	<0.001
Creatinine (mg/dL)	0.91 (0.76–1.11)	0.94 (0.79–1.12)	0.90 (0.74–1.09)	0.001
Etiology				
Alcohol, *n* (%)	428 (20.2)	164 (18.4)	264 (21.5)	0.083
HBV, *n* (%)	1073 (50.7)	479 (53.9)	594 (48.4)	0.014
HCV, *n* (%)	847 (40.0)	299 (33.6)	548 (44.7)	<0.001
Diabetes mellitus, *n* (%)	781 (36.9)	303 (34.1)	478 (39.0)	0.020
Child–Pugh score	5 (5–6)	5 (5–5)	5 (5–6)	<0.001
Class A/B, *n* (%)	1808/308 (85.4/14.6)	817/72 (91.9/8.1)	991/236 (80.8/19.2)	<0.001
FIB-4	3.22 (1.97–5.72)	2.31 (1.51–3.74)	4.12 (2.59–7.07)	<0.001
ALBI score	−2.66 (−3.02– −2.20)	−2.85 (−3.12–−2.45)	−2.52 (−2.88–−2.08)	<0.001
BCLC stage A/B/C, *n* (%)	968/590/558 (45.7/27.9/26.4)	401/272/216 (45.1/30.6/24.3)	567/318/342 (46.2/25.9/27.9)	0.633
AFP (ng/mL)	30.24 (5.70–491.17)	22.97 (4.44–399.73)	35.61 (6.95–572.83)	<0.001
AFP ≥ 400 ng/mL	564 (26.7)	222 (25.0)	342 (27.9)	0.136
Max. tumor size (cm)	3.8 (2.2–7.5)	4.5 (2.5–8.2)	3.4 (2.0–6.8)	<0.001
Tumor volume > 50%, *n* (%)	302 (14.3)	134 (15.1)	168 (13.7)	0.370
MVI, *n* (%)	427 (20.2)	147 (16.5)	280 (22.8)	<0.001
MELD score	8 (7–10)	8 (7–10)	9 (7–11)	<0.001
CLIP score	1 (0–2)	1 (0–2)	1 (0–2)	<0.001
Therapy				<0.001
Surgical resection, *n* (%)	629 (29.7)	440 (49.5)	189 (15.4)	
RFA, *n* (%)	413 (19.5)	106 (11.9)	307 (25.0)	
TACE, *n* (%)	799 (37.8)	250 (28.1)	549 (44.7)	
Systemic therapy, *n* (%)	159 (7.5)	62 (7.0)	97 (7.9)	
Radiotherapy, *n* (%)	70 (3.3)	17 (1.9)	53 (4.3)	
TACE + radiotherapy, *n* (%)	46 (2.2)	14 (1.6)	32 (2.6)	
Overall survival (months) ^1^	44.94 (39.54–50.34)	Not reached	35.48 (30.72–40.24)	<0.001

Data are presented as medians (first quartile to third quartile). ^1^ Data are presented as medians (95% confidence interval). Abbreviations: AFP, α-fetoprotein; ALBI, albumin–bilirubin; ALT, alanine aminotransferase; AST, aspartate aminotransferase; BCLC, Barcelona Clinic Liver Cancer; CLIP, Cancer of the Liver Italian Program; CI, confidence interval; FIB-4, fibrosis-4; HBV, hepatitis B virus; HCV, hepatitis C virus; INR, international normalized ratio; IQR, interquartile range; MELD, model for end-stage liver disease; MVI, macrovascular invasion; RFA, radiofrequency ablation; TACE, transarterial chemoembolization.

**Table 2 cancers-14-05083-t002:** Comparisons and different cutoffs for the ALBI grades in the original study and this study.

Variables	Original [6] (*n* = 2599 ^3^)			Eligible for Investigation (*n* = 2743)			Enrolled Patients (*n* = 2116)		
Age (years)	67 (61–72)			64 (56–72)			64 (56–72)		
Sex (male), *n* (%)	1863 (71.7)			1973 (71.9)			1513 (71.5)		
Total bilirubin (mg/dL)	0.9 (0.6–1.3)			1.0 (0.7–1.5)			0.9 (0.6–1.2)		
Albumin (g/dL)	3.5 (3.1–3.9)			3.9 (3.3–4.3)			4.0 (3.6–4.4)		
Child–Pugh A/B/C, *n* (%)	1743/684/172 (67.1/26.3/6.6)			2026/540/175 (73.9/19.7/6.4)			1808/308/0 (85.4/14.6/0)		
MVI	365 (14.1)			674 (24.6)			427 (20.2)		
Survival (months) ^1^	47.2 ^4^			33.41 (29.61–37.21)			44.94 (39.54–50.34)		
ALBI cutoff ^2^	Values	Se (%)	Sp (%)	Values	Se (%)	Sp (%)	Values	Se (%)	Sp (%)
25 percentile	−2.60	NA	NA	−2.94	85.96	37.04	−3.02	85.85	34.10
80 percentile				−1.79	28.49	89.11	−2.08	29.85	90.10
90 percentile	−1.39	NA	NA	−1.38	13.83	94.40			

Data are presented as medians (first quartile to third quartile). ^1^ Data are presented as medians (95% confidence interval). ^2^ Patients were classified into three groups by the linear predictor (at the 25th and 90th percentiles in the original study [6] and 25th and 80th percentiles in this study). ^3^ The Japanese training set [6]. ^4^ Patients in the Japanese training set had the longest median survival. The survival durations were 26.0 and 7.2 months in the Spanish and Chinese cohorts, respectively. Abbreviations: ALBI, albumin–bilirubin; MVI, macrovascular invasion; Se, sensitivity; Sp, specificity.

**Table 3 cancers-14-05083-t003:** Univariate and multivariate Cox regression analyses of the factors associated with the overall survival in all patients.

Variable	Univariate Analysis		Multivariable Analysis 1		Multivariable Analysis 2	
	HR (95% CI)	*p*-Value	HR (95% CI)	*p*-Value	HR (95% CI)	*p*-Value
Age (years)	0.996 (0.991–1.001)	0.140				
Sex: male vs. female	1.180 (1.027–1.355)	0.019				
DM: yes vs. no	0.999 (0.880–1.133)	0.985				
Platelet count (×10^9^/L) ≥100 vs. <100	0.905 (0.782–1.049)	0.186				
ALT (U/L) > 40 vs. ≤40	1.394 (1.231–1.579)	<0.001				
AFP (ng/mL) ≥ 400 vs. <400	3.094 (2.726–3.511)	<0.001	2.301 (2.009–2.635)	<0.001	2.186 (1.909–2.503)	<0.001
Tumor size (cm)	1.034 (1.030–1.038)	<0.001	1.016 (1.010–1.021)	<0.001	1.020 (1.014–1.026)	<0.001
MVI: yes vs. no	4.819 (4.216–5.508)	<0.001	2.924 (2.514–3.401)	<0.001	2.941 (2.526–3.423)	<0.001
ALBI grade 2 vs. 1	2.838 (2.493–3.230)	<0.001	2.370 (2.074–2.708)	<0.001		
ALBI grade 3 vs. 1	4.092 (3.061–5.471)	<0.001	4.196 (3.118–5.647)	<0.001		
ALBI grade 3 vs. 2	1.442 (1.086–1.915)	0.011	1.771 (1.323–2.370)	<0.001		
mALBI grade 2 vs. 1	2.063 (1.724–2.469)	<0.001			1.881 (1.568–2.257)	<0.001
mALBI grade 3 vs. 1	4.973 (4.088–6.051)	<0.001			4.001 (3.273–4.889)	<0.001
mALBI grade 3 vs. 2	2.411 (2.096–2.773)	<0.001			2.126 (1.844–2.452)	<0.001

Abbreviations: AFP, α-fetoprotein; ALBI, albumin–bilirubin; ALT, alanine aminotransferase; CI, confidence interval; DM, diabetes mellitus; HR, hazard ratio; mALBI, modified albumin–bilirubin; MVI, macrovascular invasion.

**Table 4 cancers-14-05083-t004:** Univariate and multivariate Cox regression analyses of the factors associated with the overall survival in patients receiving RFA and surgical resection.

Variable	Univariate Analysis		Multivariable Analysis 1		Multivariable Analysis 2	
	HR (95% CI)	*p*-Value	HR (95% CI)	*p*-Value	HR (95% CI)	*p*-Value
Age (years)	1.008 (0.997–1.019)	0.137				
Sex: male vs. female	1.142 (0.876–1.490)	0.327				
DM: yes vs. no	1.205 (0.950–1.527)	0.124				
Platelet count (×10^9^/L) ≥100 vs. <100	0.492 (0.380–0.636)	<0.001	0.504 (0.372–0.682)	<0.001	0.483 (0.357–0.654)	<0.001
ALT (U/L) > 40 vs. ≤40	1.141 (0.902–1.444)	0.271				
AFP (ng/mL) ≥ 400 vs. <400	2.602 (1.975–3.427)	<0.001	2.155 (1.587–2.926)	<0.001	2.257 (1.671–3.049)	<0.001
Tumor size (cm)	1.112 (1.081–1.143)	<0.001	1.095 (1.059–1.132)	<0.001	1.101 (1.066–1.137)	<0.001
MVI: yes vs. no	2.493 (1.564–3.973)	<0.001	1.783 (1.096–2.901)	<0.001	1.652 (1.014–2.692)	0.044
ALBI grade 2 vs. 1	2.414 (1.895–3.076)	<0.001	1.955 (1.503–2.543)	<0.001		
ALBI grade 3 vs. 1	4.315 (2.338–7.962)	<0.001	3.467 (1.789–6.717)	<0.001		
ALBI grade 3 vs. 2	1.787 (0.963–3.187)	0.066	1.773 (0.923–3.404)	0.085		
mALBI grade 2 vs. 1	1.747 (1.317–2.316)	<0.001			1.531 (1.146–2.046)	0.004
mALBI grade 3 vs. 1	3.669 (2.517–5.349)	<0.001			2.764 (1.817–4.204)	<0.001
mALBI grade 3 vs. 2	2.100 (1.511–2.920)	<0.001			1.805 (1.268–2.570)	0.001

Abbreviations: AFP, α-fetoprotein; ALBI, albumin–bilirubin; ALT, alanine aminotransferase; CI, confidence interval; DM, diabetes mellitus; HR, hazard ratio; mALBI, modified albumin–bilirubin; MVI, macrovascular invasion; RFA, radiofrequency ablation.

## Data Availability

The data presented in this study are available on request from the corresponding author.

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
