# Peer review of "Modified Albumin–Bilirubin Model for Stratifying Survival in Patients with Hepatocellular Carcinoma Receiving Anticancer Therapy"

_cancers, 2022, doi:10.3390/cancers14205083_

Round 1
Reviewer 1 Report
The author reported that new modified ALBI grade ;”ALBI score ≤-3.02 for mALBI grade 1, an ALBI score >-3.02 to ≤-2.08 for mALBI grade 2, and an ALBI score >-2.08 for mALBI grade 3” for HCC patients. This report is an interesting perspective. However, there were serious problems in this manuscript.
Currently, HCC was diagnosed at the advanced HCC stage, and treatment strategies are particularly important in these populations. Therefore, the original ALBI system divides ALBI grade 2 into modified ALBI 2a or modified ALBI 2b. I think ALBI garde2 in this study has a very wide range from -3.02 to -2.08. Moreover, patients who receive systemic therapy often fall within these ranges. Thus, the author should analyze using modified ALBI grade 2a or 2b, not the original ALBI grade 2. Or can the newly modified ALBI grade 2 in this study be further subdivided statistically?
Reviewer 2 Report
The article by Cheng-Yuan Peng et al, entitled ‘’ Modified albumin–bilirubin model for stratifying survival in patients with hepatocellular carcinoma receiving anticancer therapy “describes the mALBI model which can differentiate between patients with early, intermediate, or advanced HCC who received anticancer therapy. The overall manuscript is well written, the study will certainly expand our knowledge on albumin-based models in cancer therapy. There are no major points for this work. with the incorporation of the following minor change, I would enthusiastically endorse the publication of this article in Cancers.
Please improve the quality of the figures, especially the font type size to make them more legible.
Round 2
Reviewer 1 Report
This submitted paper has been appropriately refined according to the reviewer's advice.